# Toxicity of Orthodontic Brackets Examined by Single Cell Tracking

**DOI:** 10.3390/toxics10080460

**Published:** 2022-08-08

**Authors:** Morgan Wishney, Swarna Mahadevan, James Anthony Cornwell, Tom Savage, Nick Proschogo, M. Ali Darendeliler, Hans Zoellner

**Affiliations:** 1Discipline of Orthodontics, Sydney Dental School, Faculty of Medicine and Health, University of Sydney, Sydney Dental Hospital, Surry Hills, NSW 2010, Australia; 2The Cellular and Molecular Pathology Research Unit, Oral Pathology and Oral Medicine, Sydney Dental School, Faculty of Medicine and Health, The University of Sydney, Westmead Centre for Oral Health, Westmead Hospital, Westmead, NSW 2145, Australia; 3Laboratory of Cancer Biology and Genetics, Center for Cancer Research, National Cancer Institute, Bethesda, MD 20892, USA; 4School of Geosciences, Faculty of Science, The University of Sydney, Camperdown, NSW 2006, Australia; 5School of Chemistry, Faculty of Science, The University of Sydney, Camperdown, NSW 2006, Australia; 6Biomedical Engineering, Faculty of Engineering, The University of Sydney, Camperdown, NSW 2006, Australia; 7Graduate School of Biomedical Engineering, University of NSW, Kensington, NSW 2052, Australia; 8Strongarch Pty Ltd., Pennant Hills, NSW 2120, Australia

**Keywords:** single cell tracking, cytotoxicity, orthodontic brackets, material toxicity, sister cells

## Abstract

Subtle toxic effects may be masked in traditional assays that average or summate the response of thousands of cells. We overcome this by using the recent method of single cell tracking in time-lapse recordings. This follows the fate and behavior of individual cells and their progeny and provides unambiguous results for multiple simultaneous biological responses. Further, single cell tracking permits correlation between progeny relationships and cell behavior that is not otherwise possible, including disruption by toxins and toxicants of similarity between paired sister cells. Notably, single cell tracking seems not to have been previously used to study biomaterials toxicity. The culture medium was pre-conditioned by 79 days incubation with orthodontic brackets from seven separate commercial sources. Metal levels were determined by Inductively Coupled Plasma Mass Spectrometry. Metal levels varied amongst conditioned media, with elevated Cr, Mn, Ni, and Cu and often Mo, Pb, Zn, Pd, and Ag were occasionally found. The effect on human dermal fibroblasts was determined by single cell tracking. All bracket-conditioned media reduced cell division (*p* < 0.05), while some reduced cell migration (*p* < 0.05). Most bracket-conditioned media increased the rate of asynchronous sister cell division (*p* < 0.05), a seemingly novel measure for toxicity. No clear effect on cell morphology was seen. We conclude that orthodontic brackets have cytotoxic effects, and that single cell tracking is effective for the study of subtle biomaterials cytotoxicity.

## 1. Introduction

Dental materials are bathed in saliva that is swallowed for the duration of their use, and because use may extend over many years or even the lifetime of the individual, it is reasonable to consider if minor or subtle responses to these materials might have ultimately significant long-term health effects. Orthodontic appliances contain a variety of metal alloys and may be in the mouth for several years. Corrosion of appliances in the mouth is described with the release of metal constituents, including: Ni, Cr, Fr, Co, Cd, Cu, and Mn [1,2,3]. There is interest in the potential biological effects from exposure to these metals [2,4,5]. During orthodontic treatment, Ni is increased in urine, while Ni and Cr are increased in hair [6,7]. Exfoliative cytology of oral mucosa has demonstrated bioaccumulation of metals, with concomitant DNA damage and reduced cell viability [8]. Importantly, metals leached from orthodontic appliances have been detected in saliva and dental plaque [1,9], although this is not consistent across studies [10,11]. Related to this, metals have been recovered from simulated saliva soaking some facemasks [12]. Although the overall load of metal ions released from orthodontic appliances is considered to be below toxic levels, the effects of prolonged exposures at levels not immediately recognized as toxic remain essentially unknown [2].

Separately, concern emerges on the biocompatibility and potential toxicity of ‘generic’ biomedical devices, which are relatively cheap and often manufactured without regulatory oversight [13]. ‘Generic’ orthodontic brackets are widely available online at a fraction of the cost of ‘proprietary’ orthodontic brackets from recognized distributors. We recently reported high levels of Cu and Pb with known toxicity in some generic orthodontic brackets [14], and the current study extends this to evaluate cytotoxicity.

Inductively Coupled Plasma Mass Spectrometry (ICP-MS) is an exquisitely sensitive method for the detection of very low concentrations of elements relevant to toxicology in biological and other samples [15]. More importantly for the current study, ICP-MS is well accepted to evaluate metal levels in oral tissues and saliva, as well as the leaching of metals from relevant materials [11,12,16,17]. Cell death and altered proliferation are important markers for cytotoxicity, and most relevant bioassays entail measurements of global changes in populations of cultured mammalian cells [18]. However, it has recently become apparent that results from such assays may be misleading because different sub-populations of cultured cells may respond to challenge in different ways [19]. For example, an increase in cell number determined by cell counts may reasonably be interpreted as evidence for cell proliferation. This would mask, however, cell death that might be occurring at a rate lower than that of proliferation. Even worse is that while a reduction in cell number would suggest cell death in absence of cell proliferation, results of a cell counting assay cannot exclude brisk cell proliferation, overwhelmed by even more rapid cell death, and such proliferation would act as a tumor promotor [20].

These difficulties inherent to ‘pooled cell assays’ can be overcome by using the method of ‘single cell tracking’, in which the fate of individual cells is followed and quantitated from time-lapse recordings [21]. Notably, single cell tracking permits a highly nuanced measurement of a range of cellular responses in individual cells, including: morphology, cell migration, cell death, and cell division [22,23]. By providing data on multiple simultaneous cellular responses, single cell tracking obviates need to perform multiple separate assays and has the further benefit of results often being less ambiguous.

Importantly, single cell tracking has demonstrated that sister cells derived by the division of a single mother cell, more closely resemble each other than they do either the mother cell or their own progeny [22,23,24,25]. In particular, this includes the time between birth of cells and their division, so that sister cells typically show striking synchronization of cell division and cell fate. Similar, albeit less pronounced similarity and synchronization, is seen amongst ‘cousins’ in cell lineages [22,24,25]. Divergence of sister cells from one another thus provides a further potential subtle new measure of cytotoxicity.

Cell morphology can undergo significant changes in cell culture, depending on the environment [23,26,27], and so provides a further potential measure for subtle cytotoxicity. While measurement of the cell surface area requires complex high-resolution analysis of the three dimensional cell structure, the area of the cell profile presented to the microscope can be readily determined from planar images, and we define this as the ’cell profile area’. The cell profile area and perimeter are readily measured in single cell tracking recordings, and can be used to determine cell circularity by the formula: Cell circularity = 4pi(Cell profile area)/(Cell circumference)^2^, where a result of 1 is given for a perfect circle and results approach 0 with increasing cell elongation or irregularity [23]. Cell morphology measured by the cell profile area and cell circularity thus provides further potential measure for subtle cytotoxic effects that can be extracted from single cell tracking recordings. From this, single cell tracking provides an opportunity for highly sensitive evaluation of toxicity to mammalian cells. This approach has been used to study the toxic effects of a marine toxin [28] and carcinogen and chemotherapeutics [22,29,30,31,32,33], but we find no literature exploiting the method to examine the potentially subtle toxicity of biomaterials. We here describe the application of single cell tracking to investigate the possible toxicity of extracts from generic and propriety orthodontic brackets on human dermal fibroblasts (HDF).

## 2. Materials and Methods

### 2.1. Materials

#### 2.1.1. Orthodontic Brackets and Bracket Sample Coding

Five sets of generic orthodontic brackets were from www.ebay.com.au purchased on 14 February 2017, ranging in price from AU$2.99 to AU$65.06 per set with postage. These were designated with codes ‘G1’ to ‘G5’, in reducing order of cost. Proprietary brackets were from American Orthodontics (Master Series, Sheboygan, WI, USA), and Rocky Mountain Orthodontics (FLI^®^ Twins, Denver, CO, USA), and these were designated with codes ‘P1’ and ‘P2’ respectively.

#### 2.1.2. Cell Culture Materials and Reagents

Dulbecco’s Modified Eagle Medium (DMEM), bovine calf serum (BCS), glutamine, and the antibiotics penicillin, stremptomycin, and amphotericin B, were from Gibco, Life Technologies (Carlsbad, CA, USA). HDF were from the American Type Culture Collection (Manassas, VA, USA). In addition, 24 well black-bottomed plates were from (Ibidi, Gräfelfing, Germany). All remaining reagents were from Thermo Fisher Scientific (Waltham, MA, USA).

### 2.2. Preparation of Bracket-Conditioned Media

All brackets for the incisors, canines, and premolars in quadrants 2, 3, and 4 were used from all bracket sets for preparation of the conditioned media. Brackets from each set were weighed in individual 50 mL polypropylene centrifuge tubes before cleaning by: triple rinsing in 100% ethanol; air drying; and autoclave sterilization. DMEM with the antibiotics Penicillin 100 U/mL, Streptomycin 100 µg/m, and Amphotericin B 2.5 µg/mL was added to each tube to achieve a bracket weight to volume ratio of 50 mg bracket per ml [34]. The control medium comprised DMEM with antibiotics only. Tubes with brackets and the control medium were sealed with parafilm and agitated on a rolling platform for 79 days at RT. Conditioned and control media were stored frozen prior to experimentation, while 5 mL was used for elemental analysis.

### 2.3. Metal Analysis of Conditioned Media

Media in triplicate was diluted 1:8 in ultra-pure water using a Hamilton microlab 600 autodilutor before ICP-MS using a Perkin Elmer Nexion 300X (Perking Elmer, Waltham, MA, USA) with Cetac ASX-520 autosampler (Teledyne CETAC, Omaha, NE, USA). Analysis was for metals previously established as being present including: ^52^Cr, ^53^Cr, ^55^Mn, ^57^Fe, ^60^Ni, ^63^Cu, ^98^Mo, and Total Pb (sum of ^206^Pb, ^207^Pb, ^208^Pb) [14]. Several other metals of interest, including ^66^Zn, ^59^Co, ^107^Ag, and ^104^Pd were also quantified. With the exception of lead (which was run in standard mode), all metals were analyzed using the Kinetic Energy Discrimination mode with helium gas as the collision gas (4 L/min) to remove molecular interferences. All elements were run with a 0.5 s integration time. The instrumentation was tested daily against the daily performance test solution provided by the manufacturer to ensure that the sensitivity of the instrument and appropriate nebulizer settings, torch position and quadrupole ion deflector settings were automatically selected for reliable results. High Purity Standards (North Charleston, SC, USA) provided external calibration standards. The standards were provided as a mixture obtained from certified reference materials, except for Mo and Pd which were provided as individual standards as certified reference materials. Internal standards containing ^45^Sc (200 ppb), ^103^Rh (10 ppb), and ^193^Ir (10 ppb) were aspirated and mixed before the sampling chamber using a mixing box to control for matrix effects and plasma energy instabilities (Plasma Energy 1.5 kW). Standard curves for isotopes tested are shown in Appendix A) and had coefficients of correlation that approximated 1 with a maximum divergence from unity of 0.001108.

### 2.4. Cell Culture

HDF were cultured on gelatin (0.1%, Phosphate Buffered Saline, PBS) and in DMEM with BCS (15%) and antibiotics. Cells were passaged at a ratio of 1 to 3 and seeded for experimentation in 24 well black-bottomed plates in passage 4 at 5 × 10^4^ cells per well. After adhesion and overnight culture, the medium was exchanged for 900 µL of experimental medium. Experimental media comprised 50% bracket-conditioned medium or control with 50% fresh culture medium. Fresh culture medium comprised DMEM with 20% BCS, antibiotics, and glutamine (29 g/mL). Phase contrast time-lapse recordings were collected using a Leica DFC365 FX camera (Leica Camera AG, Wetzlar, Hesse, Germany) at 15 min intervals over 7 days in a humidified culture chamber under CO_2_ (5%) in a Leica DM 16000B light microscope (Leica Camera AG, Wetzlar, Hesse, Germany), using x20 objective. Grids of nine contiguous sites (3 × 3) per well were monitored. Medium depletion was avoided by changing the conditioned medium as described above on day 4 of observation.

### 2.5. Single Cell Tracking

Contiguous sites in grids were stitched into single time-lapse movies, and 25 randomly selected cells from each image were tracked over 7 days culture. Single cell tracking software developed in MATLAB (Nordon cell tracking software, R Nordon, University of New South Wales, Kensington, Australia) was used to follow the fate of individual cells and their progeny [22,25]. Characteristics monitored were: cell position, cell migration, time of cell division and birth, and time of cell death. In addition, the cell profile area and cell circularity were measured for all cells on day 3.5 of the experiment, using Image J software (Wayne Rasband, National Institute of Health, Bethesda, MD, USA). 

### 2.6. Statistical Analysis

Statistical analyses were completed with SPSS software (IBM SPSS 23.0, Armonk, NY, USA) and Graphpad Prism (GraphPad Software 7.02, La Jolla, CA, USA). The binary statistic calculator available at: https://stattrek.com/online-calculator/binomial.aspx and accessed on 9 July 2019 was used to evaluate the statistical significance of differences for cell division, applying an expected probability of 0.3617 from the control. Differences in unpaired sister cell division were evaluated using the same binomial statistic calculator applying an expected probability of 0.2789 from the control. Linear regression was used to correlate metal ion concentration with the cellular variables that were measured. Distance migrated, circularity, and the cell profile area were compared using Mann–Whitney U tests, and mitosis was compared using Chi-Square tests. Statistical significance was accepted at a value of *p* < 0.05 with Bonferroni correction where appropriate.

## 3. Results

### 3.1. Metal Ions Were Released from Brackets

Metal levels in conditioned media are shown in Table 1. There was no clear relationship between metal levels in the conditioned media and bracket composition as earlier determined. The control solution had the highest level of Fe, and lowest levels for most other metals studied. Comparing levels of metals measured in the control medium with the lowest levels detected amongst individual test samples: Fe, Cu, Mo, Pb, Zn, and Pd quantities were in the same order of magnitude as the control; while Cr, Mn, and Ni were at levels one order of magnitude greater than control. Quantities of metals detected from different samples varied greatly, such that: for Fe and Pb, all test samples measured were within the same order of magnitude; for Cr, Mn, and Zn, test samples varied within one order of magnitude; and for Mo and Pd, samples varied across two orders of magnitude. Ni and Cu had the greatest variation in levels measured amongst test samples, both ranging over three orders of magnitude. Ag and Co were both unique, in that these were only detected in one of two separate bracket extracts each.

### 3.2. Conditioned Media from Some Brackets Reduced Fibroblast Proliferation

The total number of cells tracked varied amongst control and conditioned media due to differences in cell division, migration of cells out of field, obscuring of cells by over-confluence, and apoptosis. The control medium had the most cells tracked (Table 2). All bracket-conditioned media reduced the number of cell divisions, and hence, the final number of cells tracked (Table 2). Using the binomial statistical test, a significant reduction in mitosis was seen in all bracket-conditioned media relative to the control (*p* < 0.05), when the original 25 clones were expressed as a proportion of the number of sister cells. The effect on final cell density is apparent by visual inspection of Figure 1.

### 3.3. Some Bracket-Conditioned Media Reduced Fibroblast Migration

There was significant diversity amongst samples tested in reference to cell migration. Cells in the control media migrated faster compared with cells in the conditioned media from P1, P2, G1, and G4 (*p* < 0.05, Figure 2). There was no clear difference in cell migration between proprietary and generic brackets in regards to cell migration (Figure 2).

### 3.4. Bracket-Conditioned Media Had Little Effect on Cell Circularity and Cell Profile Area

In regard to the effect of bracket-conditioned media on cell morphology, only the medium conditioned with G1 changed cell circularity, and only the medium conditioned with G4 changed the cell profile area compared with the control (Figure 3). Separately, cell circularity in P2 conditioned media was different compared to that of cells cultured in media conditioned with G2, G3, G4, and G5 (*p* < 0.05, Mann–Whitney U test). Additionally, the cell profile area in G4 conditioned medium was reduced compared to all other treatment conditions studied, with the exception of the G5 conditioned medium (*p* < 0.05, Mann–Whitney U test). 

### 3.5. Loss of Paired Sister Cell Division in Bracket-Conditioned Medium

Figure 4 shows the relative percentage of sister cells that failed to divide when their paired sister divided. This was comparatively uncommon in control conditions, relative to when cells were cultured in media conditioned by most brackets (*p* < 0.05, Binomial statistic), with the exception of media conditioned by P1 and G3. 

### 3.6. Lack of Clear Association between Discrete Metal Contaminants and Cellular Responses

While clear effects were seen of conditioning medium with brackets, there was no unambiguous correlation between the concentrations of individual metals studied, and cell division, migration, circularity, cell profile area, or unpaired sister cell division.

## 4. Discussion

Single cell tracking overcomes a number of significant limitations in traditional bio-assays using pooled cultured cells [21]. Highly nuanced and subtle effects on morphology, cell division, migration, and the normal pairing of sister cell division can be detected by single cell tracking, and the current study underscores the power of this experimental strategy to investigate cytotoxicity. While single cell tracking has been used to study the response of cells to toxin, carcinogen, and therapeutics [22,29,30,31,32,33], ours seems to be the first study to apply single cell tracking to investigate cytotoxicity of a biomaterial.

In line with previous research [34], we used bracket weights to standardize our conditioned media. Consistent with others, we incubated brackets in culture media to extract metal ions from the orthodontic brackets [35,36]. The culture medium has the advantage of extracting both polar and non-polar substances, as well as supporting cell growth [37]. As recommended from a previous study, we chose an extended incubation time to maximize the release of metal ions [34]. Interestingly, the highest metal release was seen in one of the proprietary brackets (P1), which had elevated levels of Cu, Ag, and Pd. These metals are known to be present in the brazing material of this bracket [14]. This is consistent with previous reports, raising concerns about the biocompatibility of orthodontic brazing alloys [38,39]. Whilst some earlier reports did not quantify metals extracted from brackets [34,40], or examined only a limited range of metals [41], we investigated a broad range of metals using a highly sensitive quantitative technique (ICP-MS).

Our metal ion release data are consistent with earlier studies that failed to find clear correlations between bracket composition, and the metal ions released in solution [41]. The predominant metal present in the brackets was previously determined to be Fe [14]. Notably, the amount of Fe in the conditioned media remained below that in the control medium. Considering that orthodontic brackets are known to release large quantities of Fe when placed in other types of aqueous media [42], our data suggest that Fe in the medium underwent ion exchange with another metal in the bracket. Cu and Ni, which are known to be particularly labile [43], were released in disproportionately large amounts. Importantly, Cu has been shown to be one of the most cytotoxic metals released form orthodontic appliances [34,38,44]. Considering the low incidence of apoptosis observed across all treatments, we conclude that metal ions remained at levels that are sublethal for HDF.

Nevertheless, despite a lack of observed cell death, the bracket-conditioned medium clearly affected fibroblast behavior, and to a lesser extent, cell morphology. In general, there was a reduction in the motility of fibroblasts. Perhaps the most striking observation, was the high frequency of unpaired sister cell divisions in most bracket-conditioned media. Under normal conditions, it is expected that mitosis of sister cells occurs synchronously [24]. We speculate that, given both sister cells are in identical media, and have an identical history compared with one another, the observed lack of pairing may indicate the emergence of individual genetic or epigenetic lesions in individual sister cells. Confirmation of this suspicion awaits single cell sequencing and epigenetic studies. DNA injury would be consistent with separate reports on DNA damage detected in comet assays [35,45]. Unpaired sister cell division has not, to our knowledge, been previously reported as a toxic effect of any biomaterial.

Due to the novel nature of our protocol and the fact that cell lines inherently vary in their susceptibility to metal exposure [37], only limited comparison between our findings and previous research was possible. One study using light microscopy failed to find any morphological changes in gingival fibroblasts exposed to metal bracket extracts, and this may reflect the non-quantitative method used [40]. Nonetheless, here too we found no effect on cell morphology of most bracket-conditioned media, relative to the control. Largely consistent with our finding of reduced HDF cell division, is an earlier report of reduced mouse fibroblast mitosis by archwires [46]. However, we were unable to attribute any of the responses seen to changes in levels of any of the ions detected, whereas Mn appeared responsible for the reported effect on mouse fibroblasts [46]. In a separate study, however, stainless steel wires had no effect on human fibroblast proliferation [47].

Previous attempts to correlate cell culture density with metal ion concentrations have shown mixed results. At least one study found that the composition of appliances does not correlate well with cytotoxicity [44], and others have reported that Ni and Cr levels only weakly correlate with DNA damage assed by comet assays [45]. Another study implicated Mn in suppression of fibroblast proliferation [46]. A possible future direction of our research would be to explore dose–response relationships between various metals and fibroblast activity.

Contrary to expectation, proprietary and generic brackets did not differ significantly in cytotoxic effects in the current single cell tracking study. Similarly, no clear relationship was apparent between metal levels and the cellular variables measured. It is possible that other substances may have been released from the brackets which could account for the changes seen. For example, the marker paint on the brackets may contain organic compounds which affected the cells. Notably, marker paint extracted from a range of orthodontic brackets has variable cytotoxicity for fibroblasts, independent of the origin of the brackets [48].

## Figures and Tables

**Figure 1 toxics-10-00460-f001:**
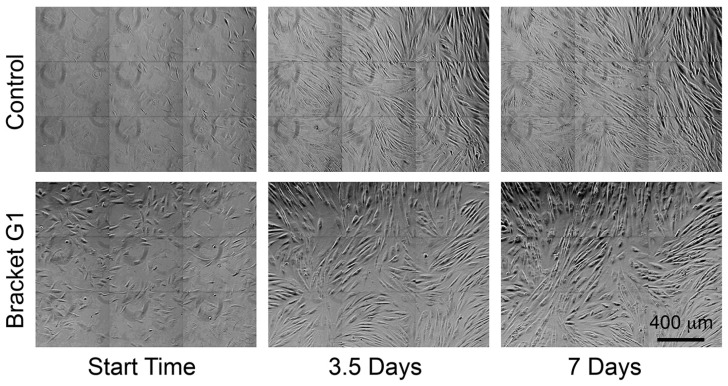
Photomicrographs of fibroblasts cultured in control conditions, as well as with culture medium conditioned with orthodontic brackets (G1) from the starting time through to 7 days. Increased cell culture density reflects cell division, and is lower in the bracket-conditioned medium compared with control cells. Cells in G1 conditioned medium also appeared more circular compared with controls. Movies of these cultures are provided in Appendix A.

**Figure 2 toxics-10-00460-f002:**
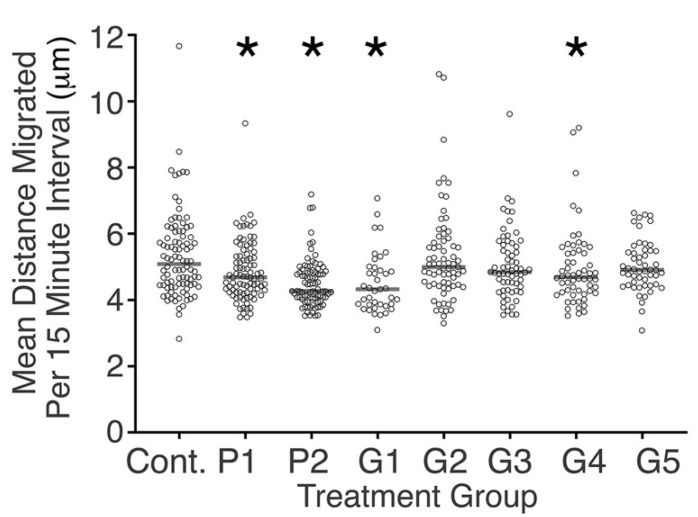
Scattergram showing the mean distance that each tracked cell migrated in intervals of 15 min. Median values for each condition studied are shown as horizontal lines. Significant differences in migration rate were observed between cells of some conditioned media and control cells (* *p* < 0.05, Mann–Whitney U test).

**Figure 3 toxics-10-00460-f003:**
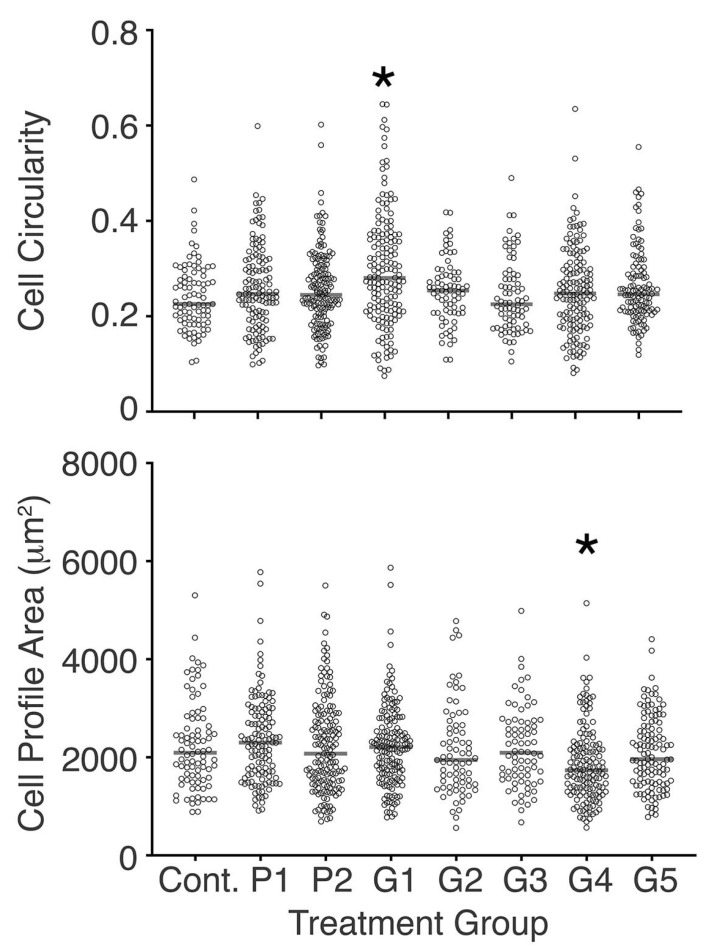
Scattergrams showing cell circularity and cell profile area on day 3.5 of culture, for all cells tracked at that time point. Median values are shown by horizontal bars. Only modest effects of conditioned media were seen, with the exception of medium conditioned by G1 and G4 where there was a significant difference compared with control cells (* *p* < 0.05, Mann–Whitney U test).

**Figure 4 toxics-10-00460-f004:**
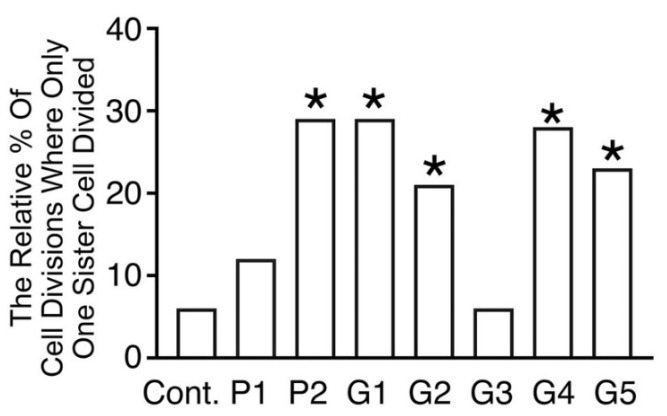
Histograms showing the relative percentage of sister cells where there was unpaired cell division. Unpaired sister cell division was comparatively uncommon in the control medium, but increased relative to the control in almost all bracket-conditioned media (* *p* < 0.05, Binomial statistic).

**Table 1 toxics-10-00460-t001:** Concentration of metal ions (Parts per billion, ppb) in each conditioned medium and control medium. Relative standard deviation percentages are indicated in brackets (*n* = 3).

	Cr	Mn	Fe	Ni	Cu	Mo	Pb	Zn	Pd	Ag	Co
Control	0.6 (11%)	1.0 (11%)	85.7 (4%)	0.8 (30%)	3.7 (54%)	0.2 (13%)	0.2 (23%)	1.6 (2%)	0.4 (17%)	0.1 (8%)	0.0 (16%)
P1	16.6 (5%)	10.2 (5%)	68.5 (11%)	65.6 (2%)	19,500.1 (1%)	1.6 (17%)	0.1 (41%)	1.7 (6%)	16.2 (4%)	3.7 (16%)	0.0 (9%)
P2	9.3 (2%)	3.9 (7%)	75.9 (9%)	10.6 (8%)	101.1 (5%)	2.8 (5%)	0.1 (26%)	1.1 (5%)	0.5 (6%)	0.0 (17%)	0.0 (9%)
G1	5.5 (3%)	7.3 (9%)	77.7 (4%)	16.0 (2%)	337.0 (2%)	5.8 (6%)	0.1 (22%)	3.6 (7%)	0.3 (4%)	0.0 (80%)	0.9 (3%)
G2	2.5 (4%)	2.2 (8%)	60.3 (14%)	7.4 (9%)	432.6 (1%)	0.7 (4%)	0.1 (36%)	1.8 (16%)	0.3 (14%)	0.0 (45%)	0.0 (19%)
G3	3.5 (3%)	1.7 (15%)	52.3 (6%)	8.7 (4%)	316.1 (2%)	3.9 (10%)	0.1 (89%)	1.2 (21%)	0.2 (2%)	0.0 (142%)	0.0 (6%)
G4	5.4 (5%)	1.7 (7%)	78.5 (9%)	1923.7 (4%)	1147.3 (2%)	22.8 (6%)	0.1 (63%)	0.8 (17%)	0.4 (10%)	0.0 (27%)	0.1 (15%)
G5	3.4 (5%)	3.9 (5%)	40.8 (1%)	6.3 (26%)	10.7 (9%)	0.2 (10%)	0.6 (4%)	1.8 (7%)	0.2 (16%)	0.0 (22%)	0.0 (10%)

Levels of most metal ions studied were significantly elevated in conditioned media.

**Table 2 toxics-10-00460-t002:** Summary of the observed fate of tracked cells.

Treatment	Total Cells Tracked	Cell Divisions	Lost Cells	Apoptosis
Control	94	34	7	1
P1	59	16	0	1
P2	55	13	1	1
G1	39	6	0	0
G2	71	23	0	0
G3	63	15	0	0
G4	58	17	0	0
G5	52	12	3	0

Twenty-five initial parent cells were tracked for all conditions studied, and cell division produced further progeny cells for tracking. Some cells were lost to tracking by either migrating out of the field of vision or becoming indistinct amongst their fellows. Occasional cells were lost via apoptosis. There was a significant reduction in the proportion of cell divisions in conditioned media from all brackets tested, relative to the control (Binomial statistic, *p* < 0.05).

## Data Availability

An Excel spreadsheet containing all cell-tracking results is provided in Appendix A.

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
