# Peer review of "Toxicity of Orthodontic Brackets Examined by Single Cell Tracking"

_toxics, 2022, doi:10.3390/toxics10080460_

Round 1

Reviewer 1 Report

In their manuscript Wishney et al. investigated the potential toxicity of orthodontic brackets by using single cell tracking in time-lapse recordings, given that minor or subtle toxic effects might be masked in traditional assays. The study is focused on the biological effects of pre-conditioned culture medium, obtained after 79 days of incubation of medium with generic and propriety orthodontic brackets, on human dermal fibroblasts. Moreover, authors detected metal levels amongst conditioned media by Inductively Coupled Plasma Mass Spectrometry. The authors demonstrated that all brackets conditioned media reduced cell division, while some reduced cell migration, but no clear effect on cell morphology was detected. Their  results indicate that orthodontic brackets have cytotoxic effects, and prompted the authors to consider the single cell tracking as an effective method for study the subtle biomaterials cytotoxicity.

1) The manuscript is clear, written in an appropriate way, is relevant for the field and presented in a well-structured manner. The question is original and well-defined, and the results provide an advancement of the current knowledge, taking into account that single cell tracking assay seems not to have been previously used to study biomaterials toxicity.

2) The methods and study design are clearly presented, appropriate for answering the research questions, the experimental design is appropriate to test the hypothesis and the experiments have appropriate controls.

3) The manuscript’s results are reproducible since sufficient details were given in the methods section to replicate the proposed experimental procedures and analysis.

4) The iconography (figures/tables/images) are appropriate, and easy to interpret and understand. Supplementary material is provided. Statements and conclusions are drawn coherent and supported by the listed citations. The cited references are relevant, and 8 out of 41 were published within the last 5 years; the bibliography does not include many self-citations.

5) The conclusions are consistent with the evidence and arguments presented, statistical tests used are appropriate and correctly reported.

6) English language is clear and understandable.

Author Response

We thank Reviewer 1 for consideration of our manuscript and the helpful suggestions for improvement.  We have addressed all of the concerns raised and believe this has significantly improved the work, while we hope that there will be agreement that the revised manuscript is now suitable for publication. 

Please note that the insertion of additional text including an updated Table 1 has disrupted the lay-out of figures and tables, but we have not shuffled figures and tables about to avoid obscuring the changes made. While the track changes document has all changes marked, the pdf provided has track changes accepted, and might be easier to read. 

COMMENTS FROM REVIEWER 1:

Although no specific points were raised by this reviewer, some general improvement was suggested.

REVIEWER 1, CONCERN 1: 

Reviewer 1 suggested that the introduction could be improved with regard to sufficient background and relevant references.

RESPONSE TO REVIEWER 1, CONCERN 1:

The following additional text has been included in the introduction and we believe that this addresses this concern.

Page 2 of the Track Changes Document, Lines 52 to 55: Insertion of the passage:  'Importantly, metals leached from orthodontic appliances have been detected in saliva and dental plaque [1,9], although this is not consistent across studies or the times sampled [10,11].  Related to this, metals have been extracted in simulated saliva from some facemasks [12].'

Page 2 of the Track Changes Document, Lines 65 to 69: Insertion of the passage:  'Inductively Coupled Plasma Mass Spectrometry (ICP-MS) is an exquisitely sensitive method for the detection of very low concentrations of elements relevant to toxicology in biological and other samples [15].  Importantly for the current study, ICP-MS is well accepted to evaluate metal levels in oral tissues and saliva, as well as the leaching of metals from relevant materials [11,12,16,17]'.

REVIEWER 1, CONCERN 2:  

Reviewer 1 suggested there could be improvement with regard to cited references being relevant to the research. 

RESPONSE TO REVIEWER 1, CONCERN 2:

We have included the below additional references as shown in the modified text, and believe this addresses the deficiency identified by this reviewer in the first submission.

Page 12 of the Track Changes Document, Lines 432 to 450:

  1. Fors, R.; Persson, M. Nickel in dental plaque and saliva in patients with and without orthodontic appliances. The European Journal of Orthodontics, 2006. 28,292-297.
  2. Lages, R.B.; Bridi, E.C.; Pérez, C.A.; Basting, R.T. Salivary levels of nickel, chromium, iron, and copper in patients treated with metal or esthetic fixed orthodontic appliances: A retrospective cohort study. Journal of Trace Elements in Medicine and Biology, 2017. 40,67-71.
  3. Petoumenou, E.; Arndt, M.; Keilig, L.; Reimann, S.; Hoederath, H.; Eliades, T.; Jäger, A.; Bourauel, C. Nickel concentration in the saliva of patients with nickel-titanium orthodontic appliances. American Journal of Orthodontics and Dentofacial Orthopedics, 2009. 135,59-65.
  4. Bussan, D.D.; Snaychuk, L.; Bartzas, G.; Douvris, C. Quantification of trace elements in surgical and KN95 face masks widely used during the SARS-COVID-19 pandemic. Sci Total Environ, 2022. 814,151924.
  5. Wilschefski, S.C.; Baxter, M.R. Inductively Coupled Plasma Mass Spectrometry: Introduction to Analytical Aspects. Clin Biochem Rev, 2019. 40,115-133.
  6. Romano, F.; Castiblanco, A.; Spadotto, F.; Di Scipio, F.; Malandrino, M.; Berta, G.N.; Aimetti, M. ICP-Mass-Spectrometry Ionic Profile of Whole Saliva in Patients with Untreated and Treated Periodontitis. Biomedicines, 2020. 8.
  7. Amr, M.A.; Helal, A.F.I. Analysis of trace elements in teeth by ICP-MS: Implications for caries. Journal of Physical Science, 2010. 21,1-12.

Reviewer 2 Report

Wishney, Zoellner and co-workers describe exciting single cell tracking and elemental analysis of human dermal fibroblasts, investigating the response of biomaterials on cellular toxicity, cell division/migration and morphology. I find this work to be highly exciting and stimulating for both bio-inorganic chemistry and analytical chemistry research fields. This work is well written and provides excellent clarity throughout the manuscript. I would please ask that the authors address some minor revisions:

Methodology - though I acknowledge the authors have cited a previous work, I would ask that the authors please provide additional clarification for ICP-MS methodology in this paper since this is key to some of their discussion; for example, selection of isotopes, specific instrumentation, internal standard concentration(s), calibration range, LOD/LOQ per analyte, etc.

In data tables, are uncertainties available to confirm whether metal ion levels were truly 'significantly' elevated? Currently unclear whether difference in values is a consequence of working close to instrument LOD/LOQ. Alternatively, I would suggest the authors report experimentally determined LOD/LOQ for each analyte in methodology (see comments above). 

Overall, this is a very well presented manuscript which provides interesting and novel insights into the interactions of dental biomaterials, and possible bioinorganic and cellular biological consequences. 

Author Response

We thank Reviewer 2 for consideration of our manuscript and the helpful suggestions for improvement.  We have addressed all of the concerns raised and believe this has significantly improved the work, while we hope that there will be agreement that the revised manuscript is now suitable for publication. 

Please note that the insertion of additional text including an updated Table 1 has disrupted the lay-out of figures and tables, but we have not shuffled figures and tables about to avoid obscuring the changes made. While the track changes document has all changes marked, the pdf provided has track changes accepted, and might be easier to read. 

REVIEWER 2, CONCERN 1:

Methodology - though I acknowledge the authors have cited a previous work, I would ask that the authors please provide additional clarification for ICP-MS methodology in this paper since this is key to some of their discussion; for example, selection of isotopes, specific instrumentation, internal standard concentration(s), calibration range, LOD/LOQ per analyte, etc.

RESPONSE TO REVEIWER 2, CONCERN 1:

We agree with this sensible suggestion, and have now significantly expanded the relevant section in the Materials and Methods detailing ICP-MS methodology.  In addition, we provide standard curves for elements detected in Supplementary Materials.  Changes to the manuscript and Supplementary Materials responding to this concern are as follows:

Pages 3 and 4 of the Track Changes Document, Lines 141` to 160: The following text in Methods 2.3 has been significantly expanded to address this concern: 'Media in triplicate was diluted 1:8 in ultra-pure water using a Hamilton microlab 600 autodilutor before ICP-MS using a PerkinElmer Nexion 300X with Cetec ASX-520 autosampler. Analysis was for metals previously established as present including: 52Cr, 53Cr, 55Mn, 57Fe, 60Ni, 63Cu, 98Mo and Total Pb (sum of 206Pb, 207Pb, 208Pb) [14]. Several other metals of interest, including 66Zn, 59Co, 107Ag and 104Pd were also quantified. With the exception of lead (which was run in standard mode), all metals were analyzed using Kinetic Energy Discrimination mode with helium gas as the collision gas (4 L/min) to remove molecular interferences. All elements were run with a 0.5 s integration time. The instrumentation was tested daily against the daily performance test solution provided by the manufacturer to ensure that the sensitivity of the instrument and appropriate nebulizer settings, torch position and quadrupole ion deflector settings were automatically selected for reliable results. High Purity Standards (North Charleton, SC, USA) provided external calibration standards. This was a mixed standard obtained from certified reference materials, except for Mo and Pd which were provided as individual standards as certified reference materials. Internal standards containing 45Sc (200 ppb), 103Rh (10 ppb) and 193Ir (10 ppb) were aspirated and mixed before the sampling chamber using a mixing box to control for matrix effects and plasma energy instabilities (Plasma Energy 1.5 kW). Standard curves for isotopes tested are shown in Supplementary Materials (Figures S1 and S2), and had coefficients of correlation that approximated 1 with a maximum divergence from unity of 0.001108.'

Page 11 of the Track Changes Document, Lines 391 to 392: Insertion of a sentence describing availability of Supplementary Figures.  'Supplementary Figures showing standard curves for ICP-MS. for all isotopes studied.'

Page 2 of the Supplementary Materials, Figures:  Figure S1. Graphs of ICP-MS standard curves for dilutions of commercially sourced standards for  206Pb, 207Pb, 208Pb, 104Pd and 107Ag, as well as calculated summated Pb for all Pb isotopes (Pb-Tot), relating counts per second (CPS) to parts per billion (ppb).  Linear relationships between the standard concentrations expected from dilutions and integrated ICPMS peaks were seen for all isotopes studied, and equations (Eqn.) for these are shown together with correlation coefficients (Cor. Coeff.). Calculated background equivalent concentrations (BEC) and detection limits (DL) for each calibrated isotope are shown. 

Page 3 of the Supplementary Materials, Figures;   Figure S2.  Graphs of ICP-MS standard curves for dilutions of commercially sourced standards for  206Pb, 207Pb, 208Pb, 104Pd and 107Ag, as well as calculated summated Pb for all Pb isotopes (Pb-Tot), relating counts per second (CPS) to parts per billion (ppb).  Linear relationships between the standard concentrations expected from dilutions and integrated ICPMS peaks were seen for all isotopes studied, and equations (Eqn.) for these are shown together with correlation coefficients (Cor. Coeff.). Calculated background equivalent concentrations (BEC) and detection limits (DL) for each calibrated isotope are shown. 

REVIEWER 2, CONCERN 2:

In data tables, are uncertainties available to confirm whether metal ion levels were truly 'significantly' elevated? Currently unclear whether difference in values is a consequence of working close to instrument LOD/LOQ. Alternatively, I would suggest the authors report experimentally determined LOD/LOQ for each analyte in methodology (see comments above). 

RESPONSE TO REVIEWER 2, CONCERN 2:

We thank Reviewer 2 for alerting us to the need to provide indication of the degree of certainty in the results shown.

As now outlined in the materials and methods (response to Concern 1 above), standard curves were generated for each isotope measured using commercially available standards, and correlation coefficients were calculated for the linear relationships observed.  These correlation coefficients closely approximated '1' and ranged from 0.998892 to 0.999985. This lends confidence to the measurements made and is addressed in the manuscript as below. Also, samples were measured in triplicate with relative percentage standard deviations reported in Table 1 (Page 6 of the Track Changes document).

Page 3 of the Track Changes Document, Line 141: Insertion of a statement that samples were evaluated in triplicate, 'Media in triplicate were ...'

Page 4 of the Track Changes Document, Lines 158 to 160: The following text has been included in Materials 2.3:   'Standard curves for isotopes tested are shown in Supplementary Materials (Figures S1 and S2), and had coefficients of correlation that approximated 1 with a maximum divergence from unity of 0.001108.'

Page 6 of the Track Changes Document Table 1 has been expanded to include the relative percentage standard deviation for triplicate samples in brackets for all elements and samples measured.

Also as mentioned in the response to Concern 1 above, Supplemental Materials now include two figures that show standard curves for all elements tested. 

Reviewer 3 Report

The article by Wishney, is of significant importance. The article does fall within the aims and scope of Toxins. The authors should add the following citation to the paper. Within the opening line the authors describe saliva and material responses. I would like to point to the following article from Bussan et al 2022. Within this article the authors simulated saliva experiments where they were able to pull out Zn, Sb and Pb. I believe this article should be pointed out for the importance of how saliva was able to extract these elements.

Bussan, Derek D., Liliya Snaychuk, Georgios Bartzas, and Chris Douvris. "Quantification of trace elements in surgical and KN95 face masks widely used during the SARS-COVID-19 pandemic." Science of The Total Environment 814 (2022): 151924.

The authors need to include the instrument parameters for the ICP-MS. Also were there any reference materials used to make sure that the ICP-MS was accurately measuring the elements of interest? I have personally seen where laboratories measure chromium only for it to be a false positive because of interferences. The quality control and assurance portion must be expanded upon. The results regarding the copper concentration of 1.95 ppm needs to be expanded upon. This is a very high concentration of copper. The authors need to be able to stand behind the data with some type of reference material(s) or quality control procedures. Please expand upon this.

If the authors are willing to expand upon what I have mentioned above I would be willing to accept the manuscript.  

Author Response

We thank Reviewer 3 for consideration of our manuscript and the helpful suggestions for improvement.  We have addressed all of the concerns raised and believe this has significantly improved the work, while we hope that there will be agreement that the revised manuscript is now suitable for publication. 

Please note that the insertion of additional text including an updated Table 1 has disrupted the lay-out of figures and tables, but we have not shuffled figures and tables about to avoid obscuring the changes made. While the track changes document has all changes marked, the pdf provided has track changes accepted, and might be easier to read. 

REVIEWER 3, CONCERN 1:

The authors should add the following citation to the paper. Within the opening line the authors describe saliva and material responses. I would like to point to the following article from Bussan et al 2022. Within this article the authors simulated saliva experiments where they were able to pull out Zn, Sb and Pb. I believe this article should be pointed out for the importance of how saliva was able to extract these elements. Bussan, Derek D., Liliya Snaychuk, Georgios Bartzas, and Chris Douvris. "Quantification of trace elements in surgical and KN95 face masks widely used during the SARS-COVID-19 pandemic." Science of The Total Environment 814 (2022): 151924.

RESPONSE TO REVIEWER 3, CONCERN 1:

We thank Reviewer 3 for bringing our attention to this interesting publication.  It is now included in the Introduction.

Page 2 of the Track Changes Document, Lines 54 to 55:  The following sentence has now been added:  'Related to this, metals have been extracted in simulated saliva from some facemasks [12}.'

REVIEWER 3, CONCERN 2:

The authors need to include the instrument parameters for the ICP-MS.

RESPONSE TO REVIEWER 3, CONCERN 2:

Reviewer 3 is correct to indicate need to provide more complete information on instrument parameters for ICP-MS.  The relevant methods section has been appreciably expanded and we believe that this addresses the identified deficiency in our first submission.

Page 3 of the Track Changes Document, Lines 141 to 152:  Text in Methods 2.3 has been re-written and expanded to now read as follows: 'Media in triplicate was diluted 1:8 in ultra-pure water using a Hamilton microlab 600 autodilutor before Inductively Coupled Plasma Mass Spectrometry (ICP-MS) (PerkinElmer Nexion 300X with Cetec ASX-520 autosampler) analysis for metals previously established as present including: 52Cr, 53Cr, 55Mn, 57Fe, 60Ni, 63Cu, 98Mo and Total Pb (sum of 206Pb, 207Pb, 208Pb) [14]. Several other metals of interest, including 66Zn, 59Co, 107Ag and 104Pd were also quantified. With the exception of lead (which was run in standard mode), all metals were analyzed using Kinetic Energy Discrimination mode with helium gas as the collision gas (4 L/min) to remove molecular interferences. All elements were run with a 0.5 s integration time. The instrumentation was tested daily against the daily performance test solution provided by the manufacturer to ensure that the sensitivity of the instrument and appropriate nebulizer settings, torch position and quadrupole ion deflector settings were automatically selected for reliable results.'

REVIEWER 3, CONCERN 3:

Also were there any reference materials used to make sure that the ICP-MS was accurately measuring the elements of interest? I have personally seen where laboratories measure chromium only for it to be a false positive because of interferences. The quality control and assurance portion must be expanded upon. The results regarding the copper concentration of 1.95 ppm needs to be expanded upon. This is a very high concentration of copper. The authors need to be able to stand behind the data with some type of reference material(s) or quality control procedures. Please expand upon this. 

RESPONSE TO REVIEWER 3, CONCERN 3:

We agree with Reviewer 3 that this was inadequately dealt with in original manuscript.  Commercially sourced reference materials were used to generate standard curves for all isotopes measured, and there are now several changes to the manuscript to address this, including provision of the ICP-MS standard curves used for quantitation in Supplemental Materials. Chromium was measured with two different isotopes to check for interferences and both isotopes selected only contain molecular interferences which would be significantly reduced using kinetic energy discrimination mode. Copper was found at this level in this sample despite 3 independent measurements of this sample on different days. This is addressed in the manuscript as outlined below.

Page 4 of the Track Changes Document, Lines 155 to 160: Internal standards containing 45Sc (200 ppb), 103Rh (10 ppb) and 193Ir (10 ppb) were aspirated and mixed before the sampling chamber using a mixing box to control for matrix effects and plasma energy instabilities (Plasma Energy 1.5 kW). Standard curves for isotopes tested are shown in Supplementary Materials (Figures S1 and S2), and had coefficients of correlation that approximated 1 with a maximum divergence from unity of 0.001108.'

Page 6 of the Track Changes Document Table 1 has been expanded to include the relative percentage standard deviation for triplicate samples in brackets for all elements and samples measured.

Page 11 of the Track Changes Document, Lines 391 to 392: Insertion of a sentence describing availability of Supplementary Figures.  Supplementary Figures showing standard curves for ICP-MS. for all isotopes studied 

Page 2 of the Supplementary Materials, Figures:  Figure S1. Graphs of ICP-MS standard curves for dilutions of commercially sourced standards for  206Pb, 207Pb, 208Pb, 104Pd and 107Ag, as well as calculated summated Pb for all Pb isotopes (Pb-Tot), relating counts per second (CPS) to parts per billion (ppb).  Linear relationships between the standard concentrations expected from dilutions and integrated ICPMS peaks were seen for all isotopes studied, and equations (Eqn.) for these are shown together with correlation coefficients (Cor. Coeff.). Calculated background equivalent concentrations (BEC) and detection limits (DL) for each calibrated isotope are shown. 

Page 3 of the Supplementary Materials, Figures;   Figure S2.  Graphs of ICP-MS standard curves for dilutions of commercially sourced standards for  206Pb, 207Pb, 208Pb, 104Pd and 107Ag, as well as calculated summated Pb for all Pb isotopes (Pb-Tot), relating counts per second (CPS) to parts per billion (ppb).  Linear relationships between the standard concentrations expected from dilutions and integrated ICPMS peaks were seen for all isotopes studied, and equations (Eqn.) for these are shown together with correlation coefficients (Cor. Coeff.). Calculated background equivalent concentrations (BEC) and detection limits (DL) for each calibrated isotope are shown. 

Round 2

Reviewer 3 Report

The authors have satisfied my requests, and the paper is now ready for publication.